



# Conceptual model of diurnal cycle of stratiform low-level clouds over southern West Africa

Fabienne LOHOU[1], Norbert KALTHOFF[2], Bianca ADLER[2], Karmen BABIĆ[2], Cheikh DIONE[1], Marie LOTHON[1], Xabier PEDRUZO-BAGAZGOITIA[3], and Maurin ZOUZOUA[4]

[1]Laboratoire d'Aérologie, Université de Toulouse, CNRS, UPS, France
[2]Institute of Meteorology and Climate Research, Karlsruhe Institute of Technology (KIT), Germany
[3]Wageningen University and Research, The Netherlands
[4]Université Felix Houphouët Boigny, Ivory Coast

*Correspondence to:* Fabienne LOHOU (fabienne.lohou@aero.obs-mip.fr)

**Abstract.**

DACCIWA (Dynamics Aerosol Chemistry Cloud Interactions in West Africa) project and the associated ground-based field experiment, which took place during the summer 2016, provided a comprehensive dataset on the low-level stratiform clouds (LLC) which develop almost every night over southern West Africa. The LLC, inaccurately represented in the climate and weather forecasts, form in the monsoon flow during the night and break up the day after, affecting considerably the radiation budget. The DACCIWA field experiment dataset supports several published studies which give an overview of the measurements during the campaign, analyze the dynamical features in which the LLC develop, and quantify the processes involved in the LLC formation. Based on the main results of these studies and new analyses, we propose in this paper a conceptual model of the diurnal cycle of the LLC over southern West Africa. Four main phases compose the diurnal cycle of the LLC. The stable and the jet phases are the two steps during which the relative humidity increases, due to the cooling of the air, until the air is saturated and the LLC form. The horizontal advection of cold air from the Guinean coast by the maritime inflow and the nocturnal low level jet (NLLJ) represents 50% of the total cooling. The remaining half is mainly due to divergence of net radiation and turbulence flux. The third step of the LLC diurnal cycle is the stratus phase which starts during the night and lasts until the onset of buoyancy driven turbulence on the following day. During the stratus phase, interactions between the LLC and NLLJ imply a modification of the wind speed vertical profile in the cloud layer, and a mixing of the subcloud layer by shear-driven turbulence below the NLLJ core. The breakup of the LLC occurs during the convective phase and can follow three different scenarios which depend on the intensity of the shear-driven turbulence observed during the night. The breakup time has a considerable impact on the energy balance of the Earth's surface and, consequently, on the depth of the convective boundary layer, which could vary by a factor of two from day-to-day.





**Keywords:** Low-level clouds, monsoon flow, maritime inflow, nocturnal low-level jet, diurnal cycle, southern West Africa





# 1 Introduction

Low-level stratiform clouds (LLC) frequently develop over southern West Africa during the monsoon season. They appear during the night, cover an extensive area and break up in the following day (Schrage and Fink, 2012; Schuster et al., 2013; Linden et al., 2015). LLC significantly affect the radiation, the diurnal cycle of the convective boundary layer (CBL), and

consequently, the regional climate (Knippertz et al., 2011; Hannak et al., 2017). However, strong biases exist between LLC representation in weather and climate models and observations, which lead to an overestimation of the solar radiation over southern West Africa by the models (Knippertz et al., 2011; Hannak et al., 2017).

Before the DACCIWA (Dynamics Aerosol Chemistry Cloud Interactions in West Africa) project (Knippertz et al., 2015), most of the knowledge about the physical processes involved in the diurnal cycle of the LLC leaned essentially on numerical

simulations, satellite images or synoptic meteorological station network measurements. LLC form in the monsoon flow, where meteorological phenomena and geographical features combine. Schrage and Fink (2012); Schuster et al. (2013); Adler et al. (2017) and Deetz et al. (2018) suggested that orographic and gravity waves lifting, horizontal cold air advection, shear-driven turbulence associated with the nocturnal low level jet (NLLJ) could all play a role in the LLC formation.

The continuous horizontal advection of cold air from the Guinean coast is maintained by the monsoon flow, but continental

southern West Africa is also impacted by the sea-breeze. During daytime, the turbulence in CBL over land decelerates the south-westerly flow, which leads to a convergence zone along a line parallel to the coast (Knippertz et al., 2017; Adler et al., 2017; Deetz et al., 2018) between the cold maritime air to the south and the warmer air in the CBL to the north. Grams et al. (2010) investigated such a quite stationary sea-breeze front during the day along the Mauritanian coast, which propagates farther inland at the end of the afternoon when the turbulence collapses. In numerical simulations, the convergence zone has

been shown to move slowly several tens of kilometers inland during the day. In the late afternoon, when turbulence vanishes in the CBL, the flow accelerates and the convergence zone moves farther and faster northwards over land (Adler et al., 2017; Deetz et al., 2018). This phenomenon is called the Gulf of Guinea Maritime Inflow (MI) in Adler et al. (2019).

Schrage and Fink (2012) and Schuster et al. (2013) suggested a dominant role played by the NLLJ on the LLC formation over southern West Africa because it intensifies the cold air advection and generates shear-driven turbulent mixing. The NLLJ

typically forms over land at the end of the day when daytime buoyancy driven turbulence ceases and the Coriolis force predominates, accelerating the wind towards low pressure (Blackadar, 1957). Although NLLJ is observed almost every night in West Africa during the monsoon season (Parker et al., 2005; Lothon et al., 2008), this formation mechanism may not be applicable at these low latitudes where the Coriolis force is weak. Parker et al. (2005) leaned on laboratory experiment of Linden and





Simpson (1986), to suggest that when turbulence rapidly diminishes, the flow is then able to respond to the pressure-gradient force. So, when the flow at low layers is decoupled from the surface and the friction force is reduced, the NLLJ develops.

One of the objectives of the DACCIWA project was to enhance, with extensive local observations, the knowledge about the LLC diurnal cycle, and determine what are the important processes that need to be addressed in the weather and climate models

to correctly simulate the LLC. A comprehensive instrumentation was implemented at three supersites, Kumasi (Ghana), Savè (Benin) and Ile-Ife (Nigeria), from mid-June to the end of July 2016. Figure 1 shows LLC occurence and average cloud fraction during the DACCIWA field campaign reported over a 55-synoptic stations network at 0600 UTC. Only LLC with more than 4 octas are considered in the statistics. In Kumasi and Ile-Ife, a 5-6 octas LLC coverage is observed with an occurrence of 80-90 %, whereas the cloud fraction is 7-8 octas with a slightly lower occurrence (70-80 %) in Savè than in Kumasi and Ile-Ife.

Kalthoff et al. (2018) and Bessardon (2019) offer an overview of the measurements during the DACCIWA field experiment at the three sites, and detail the dataset available in the BAOBAB (Base Afrique de l'Ouest Beyond AMMA Base) data base.

The most comprehensive dataset, acquired in Savè (180 km north of the Gulf of Guinea), allowed three complementary studies on the LLC. Using the 41-day period, Dione et al. (2018) give a statistical overview and a day-to-day quantification of the LLC appearance and breakup time, and of the MI and NLLJ arrival times at Savè. The statistical characteristic of the

monsoon flow and the NLLJ are also addressed. Besides this statistical overview, the days with occurrence of density current, due to convection in the surroundings of Savè, or rain, which are common phenomenon during summer season and likely influence the LLC formation, are listed. Babić et al. (2019a) analyze the diurnal cycle of the LLC at Savè on 8 July 2016, which is representative of the typical LLC development at Savè. Babić et al. (2019a) split the diurnal cycle into 5 phases (stable, jet, stratus I, stratus II and convective phases) whose characteristics are analyzed. The terms of the relative humidity

tendency equation are quantified and show that the cooling occurring during the two first phases is the main contribution to the saturation. Heat budget terms during the different phases are then analyzed to study the processes involved in the LLC formation. Adler et al. (2019) generalize this approach to a selection of 11 IOPs out of 15. The first location of the LLC appearance in southern West Africa and the way the LLC extend horizontally during the night vary a lot from one IOP to the other, but confirm that the LLC are not advected from the Guinean coast. Adler et al. (2019) and Babić et al. (2019a) quantify

and confirm the importance of the MI and NLLJ as the main processes for the low layer cooling before the cloud formation. The shear-driven turbulence in the subcloud layer, below the NLLJ core, is also pointed out as a driver of the coupling between the LLC and the surface. Additionally, Babić et al. (2019b) compared conditions during cloudy and clear nights and find that the interplay between the onset time and strength of the NLLJ, horizontal cold-air advection and background moisture level are crucial for LLC formation during the monsoon season.





Due to the rapid and significant socioeconomic changes that are happening in the southern West Africa, a substantial increase in air pollution is expected (Knippertz et al., 2011), as well as their effect on regional climate and health. Haslett et al. (2019a) show that during the monsoon season, aerosol particles in the region are likely to grow significantly because of the high moisture in the air, meaning that climate effects from increasing pollution will be enhanced. Deetz et al. (2018) performed

highly resolved process study simulations for 2–3 July 2016 with COSMO-ART to assess the aerosol direct and indirect effect on meteorological conditions over southern West Africa. They find that MI and stratus-to-cumulus transition are highly susceptible to the aerosol direct effect, leading to a spatial shift of the MI front and a temporal shift of the stratus-to-cumulus transition with changes in the aerosol amount. On the other hand, aircraft measurements of aerosols and clouds over southern West Africa during the 2016 summer monsoon show pollution and polluted clouds across the whole region (Taylor et al., 2019;

Haslett et al., 2019b). Taylor et al. (2019) conclude that smoke from biomass burning in Central Africa is transported to West Africa, causing a polluted background which limits the effect of local pollution on cloud properties.

The objective of this paper is to draw the most important lessons learned from Adler et al. (2019), Babić et al. (2019a) and Dione et al. (2018) in order to build a conceptual model of the LLC diurnal cycle observed over southern West Africa. The LLC breakup stage and the LLC impact on radiation and CBL vertical development, not addressed in Adler et al. (2019), Babić et al.

(2019a) and Dione et al. (2018), are also analyzed in this study. Our synthetic analysis is mostly based on the Savè supersite dataset. When it is possible, according to the instrumentation availability, we also apply our approach on Kumasi and Ile-Ife dataset in order to test our conceptual model on different geographical places.

The second Section presents the three instrumented supersites of the DACCIWA campaign. The differences in instrumentation or operations between Savè and the two other supersites are emphasized to explain whether the analyses can also be

applied to the Kumasi or Ile-Ife datasets. The third Section introduces the four-phase representation of the diurnal cycle of the LLC. In a first subsection, the two phases prior to the LLC formation are described and the main processes responsible of the LLC formation are addressed and quantified. In the second subsection, the LLC and the lower troposphere characteristics and their interactions are analyzed during nocturnal and turbulent convection conditions, until the LLC breakup. Discussion and conclusion are addressed in Section 4.

## 2 Instrumented sites and data

Three supersites were implemented for the DACCIWA field campaign between 14 June and 31 July 2016: Kumasi in Ghana, Ile-Ife in Nigeria and Savè in Bénin (Kalthoff et al., 2018). The ground-based campaign consisted in a 48-day period with





continuous in-situ and remote sensing observations during which 15 IOPs were conducted at the three supersites. In addition to the continuous measurements, frequent radiosondes were released at the three sites during the IOPs (Bessardon, 2019).

The study of the diurnal cycle of the LLC was conducted using the comprehensive instrumentation at Savè site (Adler et al., 2019; Babić et al., 2019a; Dione et al., 2018). As far as possible, we apply the same approach and methodology to the two other site's datasets. The most used instruments, which support the conceptual model, are listed below.

- The frequent radiosondes were launched every 1 to 1.5 hrs, depending on the IOP, between 1700 UTC on day D and 1100 UTC on day D+1 in Savè. The temperature, relative humidity and wind vertical profiles in the low atmosphere (0 to 1.5 km) were used at Savè site to study the processes involved in the LLC formation (Adler et al., 2019; Babić et al., 2019a) and dissolution. This methodology will be partly applied to the Kumasi dataset where the radiosondes were released every 1.5 to 3 hrs during some IOPs. Unfortunately, this approach is not applicable for Ile-Ife site where the soundings with a tethered ballon were sparse (every 3 hrs at the best) and did not reach a sufficiently high level. Bessardon (2019) give the overview of all radiosondes released and tethered balloon soundings at each supersite.

- A ceilometer was implemented at Savè and Kumasi sites continuously providing the cloud base height (CBH). The LLC fraction (Adler et al., 2019) was deduced from the percentage of CBH measured below 600 m and was an option to estimate the stratus appearance and breakup times (the other option was the use of the infrared cloud camera (Dione et al., 2018).

- The three sites were instrumented with surface energy balance and meteorological surface stations, which are used to study the impact of the LLC on energy balance at surface and on CBL vertical development.

- Some images of an infrared and visible camera implemented at Savè supersite are used in this study to illustrate the evolution of the state of the cloudy layer before it breaks up. The visible image was a full sky image and the aperture angles for the IR channel were $43° \times 32°$ (which corresponds to 158 m $\times$ 114 m area at a height of 200 m).

## 3  The four main phases of the LLC diurnal cycle

The schematic representing our main understanding of the diurnal cycle of the LLC is presented in Figure 2. The diurnal cycle is decomposed into 4 main phases, which will be addressed in more details in subsections. The height, in Figure 2, is normalized by the CBH when the LLC form. The first phase, named stable phase, starts around sunset. During that phase the monsoon flow is weak and the buoyancy driven convection stops. A stable layer forms close to the surface. The stable phase



**Figure 1.** Topography (grey shading) of southern West Africa with superimposed (color scale) (a) occurrence and (b) average cloud fraction of low level clouds reported at the synoptic 55-station network at 0600 UTC during the DACCIWA field campaign (14 June-31 July 2016). Only low clouds with more than 4 octas coverage are considered in the statistics. Frequency of reports (shape of the markers) indicates the percentage of days with available data during the whole period. The three supersites, Kumasi in Ghana, Savè in Benin and Ile-Ife in Nigeria, are indicated with black circles.

duration ranges from 0 to 3 hours, depending on the IOP used to build the conceptual model (Table 1). The second phase is named the jet phase. It starts with the MI arrival at the site and is characterized by the NLLJ settlement. Stable and jet phases are key phases because the processes which take place during those periods lead to an increase of the relative humidity up to saturation. The jet phase ends when LLC form, and, depending on IOP, lasts 4 to 9 hours (Table 1). The third phase is the





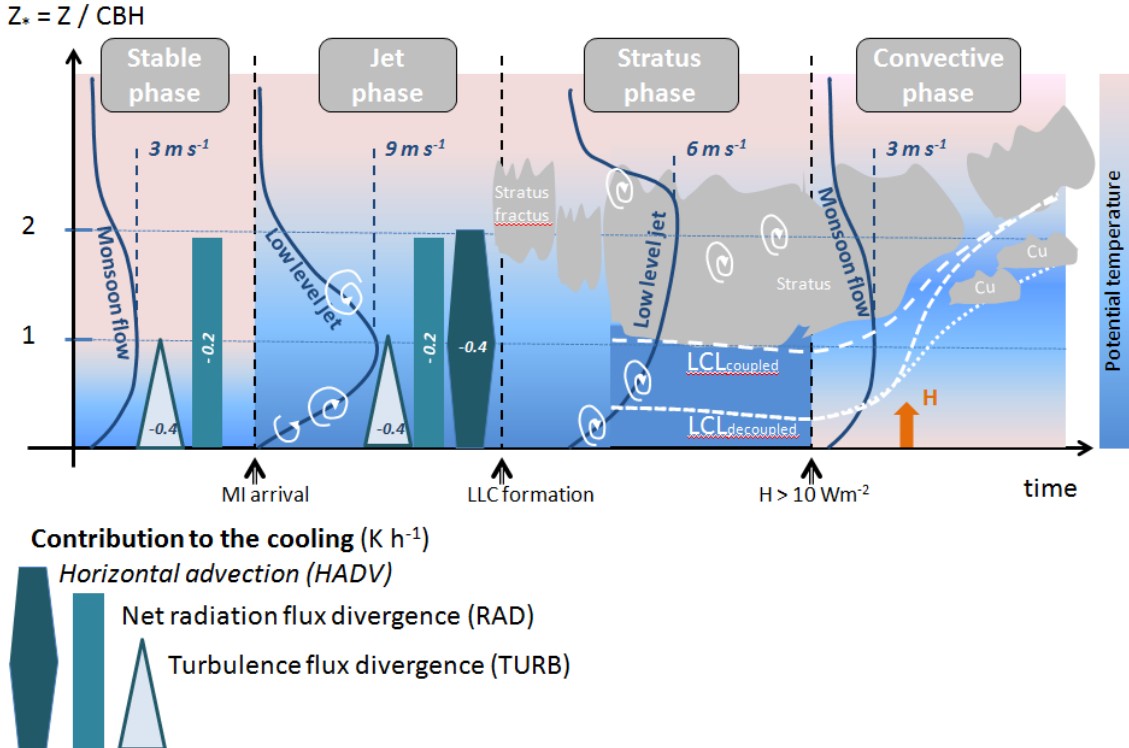

**Figure 2.** Conceptual model for the LLC diurnal cycle over southern West Africa. The height is normalized by the LLC base height (CBH) when the clouds form. The grey shades represent the LLC (stratus-fractus or stratus) or cumulus cloud. The white dashed curves indicate the lifting condensation level (LCL). The dark blue lines stand for the schematic vertical profile of the wind with indication of its maximum value for each phase. The greenish rectangles and triangles schematize and quantify the processes involved in the potential temperature tendency equation. H stands for the surface sensible heat flux and is symbolized by an orange arrow during the convective phase.

stratus phase. In some cases, a stratus fractus deck forms before the appearance of a more homogeneous deck. During this phase, strong interactions between the stratus and the sheared wind in the NLLJ exist. The stratus phase lasts between 2.5 and 16 hours (Table 1). At last, the convective phase starts with the increase of buoyancy driven turbulence and ends with the LLC breakup, between 0730 and 1500 UTC. Three different scenarios explaining the LLC breakup have been observed.

5    The characteristics of the four phases in terms of wind speed, potential temperature, relative humidity and bulk Richardson number are presented in Figure 3. The height time sections composing Figure 3 are built by (i) normalizing the height by the CBH when the stratus forms for each individual radiosounding launched at Savè supersite, (ii) setting the origin of time with a time reference appropriately chosen according to the considered phase, and (iii) hourly averaging the radiosoundings. In the left column, the origin of time is set to the MI arrival time (Dione et al., 2018): the negative and positive times stand for stable and jet phases, respectively. The jet phase stops when the stratus fractus or the stratus form. On the right column, the origin





**Table 1.** Variation of the four phases duration for the IOP included in the statistics. Variation range of MI arrival, stratus fractus and stratus onset and breakup times for the IOP included in the statistics.

|  | Duration (hour) |  | Time (UTC) |
|---|---|---|---|
| Stable phase | 0 - 3 | | |
| | | MI arrival | 1600 - 2000 |
| Jet phase | 4 - 9 | | |
| | | Stratus Fractus onset | 1900 - 0350 |
| | | Stratus onset | 2210 - 0500 |
| Stratus phase | 2.5 - 16.5 | | |
| | | Buoyancy driven turbulence onset | 0730 - 0900 |
| Convective phase | 0 - 7.5 | | |
| | | LLC breakup | 0730 - 1500 |

of time is set to the start of the convective phase defined as when surface sensible heat flux ($H$) is larger than 10 W m$^{-2}$. These panels present the conditions from the stratus appearance to the stratus breakup. The negative and positive times stand for stratus phase and the convective phase, respectively. The 9 IOPs included in the statistic for the stable and jet phases are those performed after the 30 June, when the UHF data and the microwave radiometer are available to determine the MI arrival

time (Dione et al., 2018). The other panels (stratus and convective phases) include 11 IOPs during which stratus clouds form and which are not disturbed by rain or meso-scale systems close to Savè.

### 3.1 Before cloud formation

#### 3.1.1 Description of the stable and jet phases

The MI arrival time, used here as the reference time for the stable and jet phases, has been determined by Dione et al. (2018)

using fuzzy logic method combining an increase in the wind and a decrease in the temperature. The MI arrival time ranges from 1600 UTC to 2100 UTC and is negatively correlated to the monsoon strength in the afternoon. Indeed, a strong monsoon flow favors an early MI arrival (Dione et al., 2018). When the MI arrival occurs before the establishment of the stable conditions near the surface (with negative surface sensible heat flux) the stable phase can not be defined. This situation occurs on 4 IOPs out of 9.

The stable phase is characterized by a weak monsoon flow, which persists until the MI arrival time (Fig. 3). The low layer stabilizes with an increase of the Richardson number because of the decrease of the temperature near the ground. The jet phase starts when the MI reaches the site. The NLLJ usually sets in shortly after (about 1 hr) (Dione et al., 2018). Since MI and NLLJ are both associated to the cease of the buoyancy driven turbulence, and due to the distance from the coast to Savè, both processes settle almost at the same time at Savè site (Adler et al., 2019; Babić et al., 2019a; Dione et al., 2018). They



imply a progressive increase of the wind, which reaches 10 m s$^{-1}$ at the end of the jet phase. The jet core height is located

at $Z_* = 1$. At the same time, a cooling and an increase in relative humidity occur up to $Z_* = 4$. An additional impact of the

wind increase during the jet phase is the decrease of Richardson number below the height $Z_* = 2.5$, due to the wind shear in

the layer below the jet core. Wind speed and potential temperature vertical profiles are shown schematically for stable and jet

phases in Figure 2.

### 3.1.2   Relevant processes leading to saturation

The contributions of temperature and specific humidity changes to the RH changes have been quantified for a case study (Babić

et al., 2019a) and for 11 IOPs (Adler et al., 2019) using radiosondes released at Savè site. Figure  4a shows that, along the

stable and jet phases, the cooling causes at least 80 % of the RH increase. The weak moistening actually observed during the

stable phase is almost compensated by the drying occurring during the jet phase (Adler et al., 2019) (not shown), leading to

a 20 % contribution of the moisture to the increase in RH during the two phases. Using the frequent radiosondes released at

Kumasi, the temperature and moisture contributions to the RH change have been estimated with the same method (Fig. 4b).

The results for the two sites are coherent, and confirm that the cooling is responsible for the saturation of the low layer of the

atmosphere.

In order to quantify the processes responsible for the cooling, Babić et al. (2019a) and Adler et al. (2019) used the budget

equation for the mean potential temperature ($\theta$) (Stull, 1988):

$$\underbrace{\frac{\delta\theta}{\delta t}}_{TOT} = \underbrace{-u\frac{\delta\theta}{\delta x} - v\frac{\delta\theta}{\delta y}}_{HADV} \underbrace{-w\frac{\delta\theta}{\delta z}}_{VADV} + \underbrace{\frac{1}{\rho c_p}\frac{\delta Q}{\delta z}}_{RAD} - \underbrace{\frac{1}{\rho c_p}\frac{\delta H}{\delta z}}_{TURB} - \underbrace{\frac{1}{\rho c_p}\frac{\delta L_v E}{\delta z}}_{SQ} \tag{1}$$

where $u$, $v$, $w$ are the wind components, $\rho$ the mean air density, $c_p$ the specific heat capacity of the air at constant pressure,

$H$ the sensible heat flux, $L_v$ the latent heat of water vaporization and $E$ the evaporation rate. The contributions to the tendency

of the potential temperature ($TOT$) are, the horizontal ($HADV$) and vertical ($VADV$) advection of potential temperature, the

divergence of the net radiation flux ($RAD$) and of the sensible heat flux ($TURB$) and the phase change ($SQ$). However, $SQ$

can be neglected because the heat budget analysis is applied before the cloud formation. As the vertical component of the wind

($w$) is very difficult to draw from the observations, $VADV$ is not estimated, but is expected to be small. The two remaining

terms, $TOT$ and $HADV$, are then estimated using the measurements at Kumasi and Savè supersites. $TOT$ is deduced from

the radiosonde profiles launched at the supersite. $HADV$ is calculated combining radiosondes launched at the supersite and





at three coastal stations (Abidjan, Accra and Cotonou) (see Adler et al. (2019) for details of the method). At last, the Santa

Barbara DISORT Atmospheric Radiative Transfer (SBDART) model (Ricchiazzi et al., 1998) is applied to estimate $RAD$

(see Babić et al. (2019a) for details of the method) at Savè site only. Such method could not be applied to the Kumasi dataset

because too few radiosoundings were launched during the jet phase. Additionally, applying the SBDART model to atmospheric

soundings with clouds requires some cloud characteristics (cloud top, liquid water content) which are not available at Kumasi

site. Finally, the residual term ($RES$) includes $TURB$ at Savè site (Fig. 5a) and both the $TURB$ and $RAD$ at Kumasi site

(Fig. 5b).

    Figure 5a shows a total cooling that decreases with height from -0.8 K h$^{-1}$ at surface to -0.15 K h$^{-1}$ at $Z_* = 4$. $HADV$

is also height dependent, with a vertical profile closely linked to the horizontal wind profile of the NLLJ (Adler et al., 2019;

Dione et al., 2018). $HADV$ is -0.2 K h$^{-1}$ at surface and at $Z_* = 2.5$, with a maximum cooling of -0.4 K h$^{-1}$ at the jet core

height. $RAD$ slightly increases with height from -0.15 K h$^{-1}$ at surface to -0.1 K h$^{-1}$ at $Z_* = 4$. This means that $RES$, which

is mainly the sensible heat flux divergence term at Savè, is the main contribution to the cooling close to the ground, -0.4 K

h$^{-1}$, with a decreasing effect with height. Above $Z_* = 1.5$ the sum of RAD and HADV is nearly equal to TOT. These three

processes and their respective role during the stable and jet phases are indicated in Figure 2. Adler et al. (2019) conclude, when

integrating the vertical profiles with height for Savè site, that the advection is the main cooling process, responsible for 50 % of

the cooling of the atmosphere during the stable and jet phases. Each of radiation and heat flux divergences contributes around

22% to the total cooling.

    Very similar results are obtained with the Kumasi dataset. A slightly lower total cooling is nevertheless observed at Kumasi

site. Close to surface, it is 25% lower than the one observed at Savè site, with a cooling of -0.6 Kh$^{-1}$ at Kumasi instead of -0.8

Kh$^{-1}$ at Savè. It remains 16% lower at $Z_* = 1$. Nevertheless, as for Savè, $HADV$ contributes about 50% to the total cooling

at around $Z_* = 1$ and above, while cooling near the surface is mainly caused by the sensible and radiation flux divergence.

### 3.2   From stratus onset to breakup

The two phases presented in this section are the stratus phase (stratus fractus and stratus) and the convective phase during

which the stratus breakup occurs. Dione et al. (2018), using infrared camera, shows that the stratus occurs on average 3 hours

after the NLLJ settlement at Savè site. The stratus appears after 2200 UTC and, for most of the cases, between 0000 and 0500

UTC. The breakup occurs before 1500 UTC the day after. Kalthoff et al. (2018), using net long-wave radiation from surface

measurements (available at the three supersites) as an indicator for the mean onset of LLC, shows that on average the stratus

appears earlier at Ile-Ife (2100 UTC) and Kumasi (0000 UTC) sites than at Savè (0300 UTC).


### 3.2.1 Description of the cloudy and convective phases

According to the right column in Figure 3 presenting the stratus and convective phases, the highest values of the horizontal wind speed, defining the NLLJ core, are observed between $Z_* = 1$ and $Z_* = 1.5$ at the beginning of the stratus phase. The correspondence of the stratus base height with the NLLJ core can be explained by the high cooling due to horizontal advection at this level (Fig. 5). During the stratus phase, the wind speed decreases and gets more constant with height between $Z_* = 0.5$ and $Z_* = 2$ (Fig. 3). The modification of the wind vertical profile is likely due to the turbulent mixing within the cloudy layer induced by long-wave cooling at the cloud top. Although reduced, the maximum values of the wind speed are found towards the top of the cloudy layer during the stratus phase (Adler et al., 2019; Babić et al., 2019a; Dione et al., 2018) as schematically represented in Figure 2. The potential temperature in the subcloud layer is often well mixed and two processes could contribute to this: the shear-driven turbulence below the NLLJ core and the turbulence induced by the stratus due to the radiative cooling at the cloud top. When the convective phase starts, the wind speed is almost back to monsoon flow daytime condition, that is weak wind of about 4 m s$^{-1}$. The potential temperature increases with time. The relative humidity decreases with time close to the surface and above $Z_* = 2$, which indicates a thinning of the stratus from both the base and the top. The cloudy layer rises as the CBL develops, with the typical features of a convective mixed layer: a warming of the near-neutral stratified subcloud layer and a decrease of the Richardson number.

### 3.2.2 Relevant processes linking the stratus to the surface

As shown in Figure 3, the stratus and the NLLJ interact in several ways: (i) the stratus reduces the NLLJ strength because of the turbulent mixing in the cloudy layer, (ii) the turbulence below the NLLJ core modifies the conditions in the subcloud layer. As already observed during AMMA experiment (African Monsoon Multidisciplinary Analysis, Redelsperger et al. (2006)) by Lothon et al. (2008), the NLLJ can induce dynamical turbulence down to surface, reflected by an increase of nighttime turbulent kinetic energy up to 0.6 and 0.3 m$^2$ s$^{-2}$ on average at Kumasi and Savè sites, respectively (Kalthoff et al., 2018). If the shear-driven and the cloud-top driven turbulence is strong enough, the subcloud layer is well mixed and the lifting condensation level (LCL) (Romps, 2017) will correspond to the stratus base. The stratus is considered coupled to the surface (Adler et al., 2019). On the contrary, a subcloud layer with low turbulence will be less mixed. The cloud is then decoupled from the surface, with its base higher than the LCL. This is shown in Figure 6, where the difference between the CBH and the LCL is plotted against the bulk Richardson number in the subcloud layer estimated from the radiosondes at Savè and at Kumasi. For a Richardson number below 0.1, the CBH is less than 75 m below or above the LCL. For larger values of the Richardson number, the CBH is





at least 150 m above the LCL. One can note only 4 radionsoundings for which the Richardson number is above 0.1 at Kumasi, indicating larger turbulence in the subcloud layer at Kumasi than at Savè.

### 3.2.3 Stratus breakup and boundary-layer evolution

The stratus breakup occurs during the convective phase, when the CBL develops vertically due to the heating of the surface. The

frequent radiosondes released during the convective phase are unfortunately very difficult to use to determine the CBL height because of the very weak measured gradients. An estimate of the subcloud-layer height is given by the LCL. This method is motivated by the very good accordance between this level and the cumulus cloud base height, which raises in the afternoon. In addition, this method has the advantage to continuously provide the subcloud-layer height and can easily be applied to the Kumasi and Ile-Ife sites where the radiosondes are less frequent.

The way in which the stratus layer and the surface are coupled, as discussed in the previous section, plays a determining role on the breakup. Three scenarios have been observed at Savè site, illustrated in Figure 7. For each scenario, Figure 7 presents, on the top panel, the temporal evolution, from 0000 to 1600 UTC, of the cloud base (measured by the ceilometer) and summit (measured by the cloud radar, see Adler et al. (2019) for the method) and LCL. The half-hourly standard deviation of the cloud base and of the cloud base fraction (percentage of cloud base below 1000 m a.g.l. over 30 minutes) and the difference

between cloud base and LCL are indicated on the bottom panel. A cloud fraction larger than 95 % is chosen as a criterium to determine the presence of stratus clouds above the supersite (Adler et al., 2019). Examples of visible and infrared images from the cloud camera at two distinct times illustrate the cloud coverage, and allow to make the link between the quantified parameters presented previously and the state of the sky as it can be seen by the cameras.

– Scenario 1 (Fig 7a and b): LLC coupled to the surface. It is illustrated by the 8 July case (IOP 8) (Babić et al., 2019a).

During the night, the CBH is very close to the LCL and the CBH standard deviation is very small, which indicates a very regular and constant stratus base. The visible and infrared images indicate at 0814 UTC a homogeneous stratus layer. After 0800 UTC, the LCL steadily increases to reach 1000 m a.g.l. at 1600 UTC. Until 1200 UTC, the increases of the LCL and CBH are simultaneous and identical. After that time, the cloud base fraction decrease indicates the stratus breakup. The low values of the standard deviation of the CBH during daytime, and the visible and infrared images at

1028 UTC indicate a stratus layer evolution towards a thinner cloud layer and a stratocumulus cloud. Among the 8 IOPs for which the LLC breakup has been analyzed, 3 follow this scenario. This breakup scenario is schematically presented in Figure 2 with an LCL ($LCLcoupled$) close to the CBH.


When the stratus is decoupled from the surface during the night, meaning that the stratus base is above the LCL, two scenarios can occur:

– Scenario 2 (Fig 7c and d): progressive coupling of the LLC to the surface. In this scenario, the LCL rises up to the stratus base, as illustrated with the 27 July case (IOP 14) and schematized in Figure 2. After 0800 UTC, once the stratus layer is coupled to the surface, the CBH rises with the subcloud layer height as in scenario 1. On IOP 14, the stratus breakup occurs at 1000 UTC, shown by the decrease of the cloud base fraction. In that case, the high standard deviation of the CBH indicates cumulus cloud development. The target of the ceilometer is not only the base of the cloud, but also the edge when the clouds pass over the instrument. In case of cumulus clouds, the ceilometer indicates then some very scattered measurements. Visible and infrared images illustrate the passage from the stratus layer at 0718 UTC to cumulus clouds at 1214 UTC. Two IOPs among 8 follow the scenario 2.

– Scenario 3 (Figure 7e and f): cumulus cloud formation below the stratus layer. This scenario is illustrated by IOP 11. The stratus layer is decoupled from the surface with a CBH base 150 m above the LCL during the night, increasing to 300 m at 0700 UTC. The infrared image is a mix of orange and yellow colors, which indicates colder bases than the ones observed on IOP 8 and IOP 14 early in the morning. After 0715 UTC, the ceilometer measures a quite scattered cloud base, but coupled with the surface. This indicates some cumulus cloud formation below the stratus layer. The two cloudy layers coexist until 0830 UTC as indicated by the 100 % cloud coverage. A thin stratocumulus layer persists after 0900 UTC with a very low standard deviation of the base which is at about 400 m a.g.l. When the low stratocumulus layer is broken enough, higher cloud base at around 1500 m a.g.l. are detected by the ceilometer, between 1000 and 1100 UTC. Visible and IR images at 1036 UTC illustrate this higher stratocumulus layer. Three IOPs among 8 follow the scenario 3. This breakup scenario is schematically presented in Figure 2 with an LCL ($LCL decoupled$) below the stratus base all along the convective phase and with some boundary layer cumulus formation below the stratus base.

The stratus clouds modify the energy balance at surface (SEB) and, consequently, the diurnal cycle of the CBL. In order to investigate the effect of the stratus on the SEB, the integrated flux from 0600 to 1600 UTC (indicated by $<>$) of the net radiation ($Rn$), the sensible heat flux ($H$), and the latent heat flux ($LE$) are calculated for 21 days for Savè and Kumasi supersites and 20 days for Ile-Ife.

As expected, $< Rn >$, $< H >$ and $< LE >$ are negatively correlated with the stratus breakup time with correlation coefficients below -0.63 for Savè and Kumasi supersites (Fig. 8). The variability is certainly due to the variation of the soil moisture from day-to-day which is more important at the beginning of the campaign than at the end, when frequent rain events maintain





an almost constant soil moisture. $< Rn >$ is reduced by 25% and 50% when LCL breakup occurs after 1300 UTC in Savè and Kumasi, respectively. This difference may be due to LLC macrophysical characteristics, like deeper clouds or larger liquid water path, but also to higher cloudy layers which also impact the net radiation. Unfortunately, the same plot cannot be provided for Ile-Ife because the breakup time of the stratus cannot be determined.

The impact of the SEB on the CBL vertical development is shown in Figure 9a. The CBL development is represented by the LCL at 1600 UTC at the three sites and plotted against $< H >$. The lower the $< H >$, the lower the LCL, which ranges between 300 to 1000 m a.g.l. over the three sites. The link between the CBL development and the stratus breakup time, plotted for the Savè and Kumasi sites, is also obvious (Fig. 9b) with an LCL divided by two when LLC breakup occurs after 1100 UTC compared to LCL associated with early-morning LLC breakup.

**4   Discussions and Conclusions**

The DACCIWA ground-based field campaign, conducted over southern West Africa during the 2016 monsoon season, provided a comprehensive dataset allowing the analysis of the diurnal cycle of the LLC. The most important lessons drawn from the analyzes of Savè supersite dataset (Adler et al., 2019; Babić et al., 2019a; Dione et al., 2018) and additional results on the LLC breakup scenarios and their impact on the CBL vertical development on the following day, are used to build a conceptual
model for LLC diurnal cycle and are generalized, when possible, to the two other supersites of the DACCIWA ground-based field campaign, Kumasi and Ile-Ife.

The conceptual model proposed in this study divides the LLC diurnal cycle in four main phases and addresses, for each of them, the main atmospheric processes involved.

   – The stable and jet phases constitute the period during which the cooling of the air leads to the saturation and the stratus
formation. The cooling starts at the end of the afternoon, during the stable phase, and is intensified after the MI arrival and the NLLJ settlement, during the jet phase. Three processes are in play during these phases: the horizontal advection, which contributes to 50 % of the air temperature decrease, and the net radiation and the sensible heat flux divergences, which contributes each to 22 %. In Kumasi and Savè, 94 % of the cooling prior to the LLC formation is then explained by these three processes. However, Savè and Kumasi supersites are both at the same distance from the coast and the
contributions of the three processes might change according to the distance inland, and other processes may come into play. As an example, gravity waves and orographic effects, previously pointed out in modeling studies as import factors for cloud formation, could not actually be evaluated with DACCIWA ground-based field experiment.





– The stratus phase is characterized on average by a reduction of the NLLJ in the stratus layer and often a well-mixed subcloud layer. The shear-driven turbulence associated with the NLLJ in the subcloud layer and the turbulence induced by the radiative cooling at the cloud top are most likely the relevant processes. The use of the Bulk-Richardson number in the subcloud layer takes into account the shear-driven turbulence, that is the first process, but the static stability may
result from both processes. Looking at day-to-day variability, large values of Bulk-Richardson numbers correspond to CBH higher than LCL: the LLC are then said decoupled from the surface. Close to zero Bulk-Richardson numbers are associated to CBH close to the LCL: the LLC are consider as coupled to the surface. Interestingly, only coupled cases have been observed at Kumasi during the studied IOPs, the Bulk-Richardson number in Kumasi being lower than in Savè. The coupling to the surface might play a role on the LLC characteristics, since coupled LLC seem to have a more
steady CBH during the night.

– The convective phase starts with the buoyancy driven turbulence in the CBL on the following day. The breakup of the LLC can follow three scenarios. The scenario 1 is defined when the LLC is coupled to the surface, whereas the two other scenarios can be found when the LLC are decoupled from the surface. Further analyses of the processes in play in the LLC breakup according to the scenario would be interesting. The breakup time of the LLC impacts the integrated
(from 0600 to 1600 UTC) net radiation at the surface during the following day, with 25 % reduction at Savè site and 50 % reduction at Kumasi site for late LLC breakup time compared to early LLC breakup time. The reasons for such a difference would be interesting to investigate, since it shows a possible strong disparity from one site to the other. The vertical development of the subcloud layer is directly impacted by the reduction of the available energy at surface, with a 50 % reduction of its height for a late LLC breakup time compared to an early LLC breakup time.

As shown by these results, the formation and dissipation of the LLC lean on the combination of several processes, the intensity of each of them being important in the diurnal cycle of the LLC. Thus, an accurate representation of LLC in the models needs a proper simulation of these atmospheric processes which take place, for most of them, in the lower atmospheric layer. The results presented in this study can help to highlight the weaknesses of the numerical models in the representation of the diurnal cycle of the LLC.

*Author contributions.* N

. Kalthoff, F. Lohou, M. Lothon, and B. Adler designed the DACCIWA ground-based field experiment. N. Kalthoff, F. Lohou, M. Lothon, B. Adler, C. Dione, and X. Pedruzo-Bagazgoitia performed the measurements at Savè site, processed and analyzed





the data. K. Babić and M. Zouzoua processed and analyzed the data. F. Lohou prepared the manuscript with contributions from all co-authors.

*Data availability.* T

he data used in this study are available on the BAOBAB database (Derrien et al., 2016; Handwerker et al., 2016; Kohler
5    et al., 2016; Wieser et al., 2016).

*Competing interests.* T

he authors declare that they have no conflict of interest.

*Acknowledgements.* The DACCIWA project has received funding from the European Union Seventh Framework Programme (FP7/2007-2013) under grant agreement no. 603502. The authors thank also Laboratoire d'Aérologie, Université de Toulouse, CNRS, UPS, France
10    and KIT (Karlsruhe Institute of Technology) and UPS (Université Toulouse) for helping to install the equipment as well as the people from INRAB in Savè for allowing the equipment on their ground. We thank the Aeris data infrastructure for providing access to the data used in this study.



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





**Figure 3.** Mean height-time sections for, from top to bottom, wind speed (ms$^{-1}$), potential temperature (K), relative humidity (%) and bulk Richardson number (color scale and isolines). These height-time sections are built by (i) normalizing the height by the CBH when the stratus forms for each individual radiosounding, (ii) setting the origin of time with a time reference appropriately chosen according to the considered phase, and (iii) hourly averaging the radiosoundings. Red numbers indicate the number of radio soundings available for the hourly-average. The origin of time is set to (left column) MI arrival time (negative and positive times standing for stable and jet phases, respectively), and (right column) start of the convective phase ($H > 10$ W m$^{-2}$) (negative and positive times standing for stratus phase and convective phase, respectively





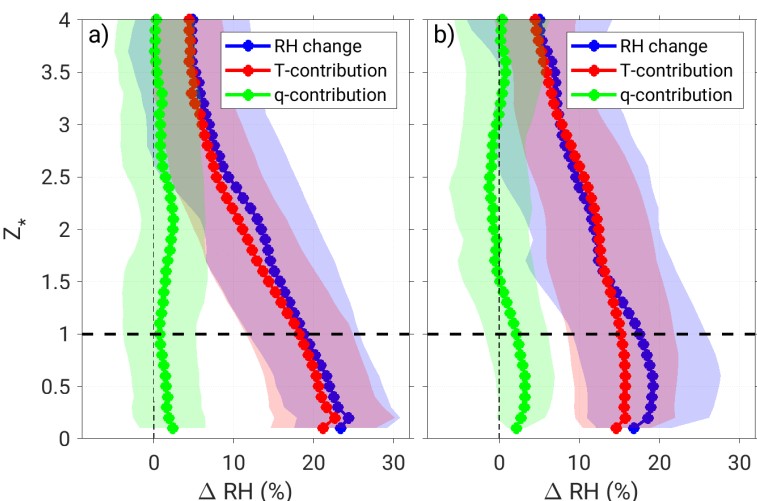

**Figure 4.** Specific humidity (q) and temperature (T) mean contributions to the mean total change in relative humidity (RH) at (a) Savè and (b) Kumasi averaged for all available IOPs. The shading indicates the standard deviation.

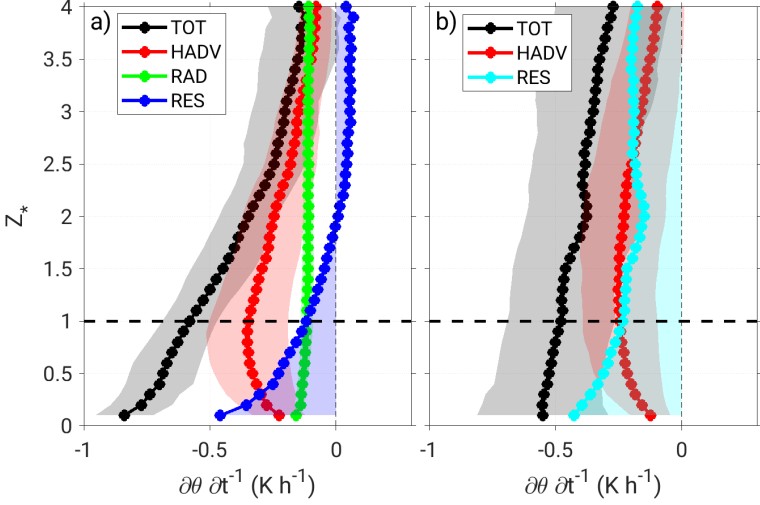

**Figure 5.** Vertical profiles of total cooling rate ($TOT$) and contribution estimates of horizontal advection ($HADV$) and net radiation flux divergence ($RAD$) terms for the Savè site during the stable and jet phases (a). Vertical profiles of $TOT$ and $HADV$ for the Kumasi site during the stable and jet phases (b). The residual term $RES$ includes the turbulent flux divergence at Savè site and both the net radiation and turbulent flux divergences at Kumasi site. The shading indicates the standard deviation.

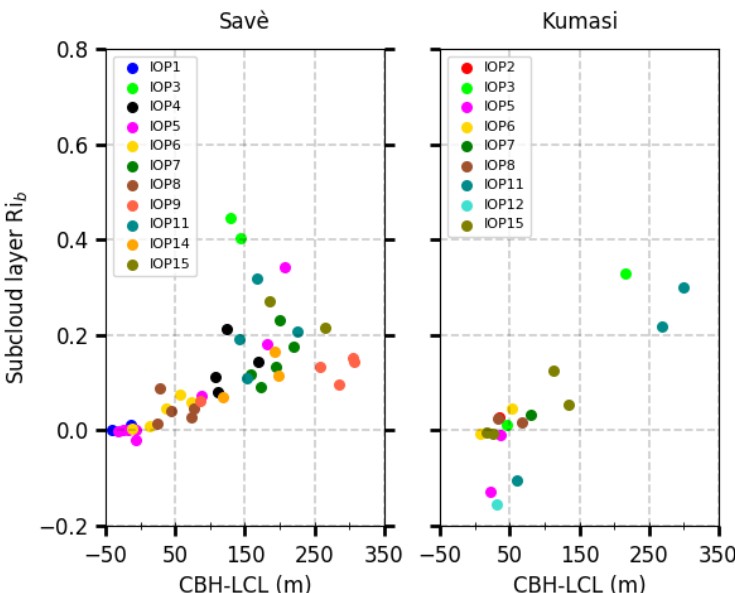

**Figure 6.** Bulk Richardson number ($Ri_b$) in the subcloud layer against height difference between cloud base height (CBH) and lifting condensation level (LCL) at Savè and Kumasi. Colors stand for the different IOPs.

**Figure 7.** (a) 8 July (IOP 8), (c) 27 July (IOP 14) and (e) 18 July (IOP 11) illustrate the scenario 1, 2 and 3, respectively. The temporal evolution of the cloud base (red dots) and summit (grey dots) and the LCL with its incertitude (black line with grey shading) are presented on the top panel. The LCL incertitude is based on the incertitudes of the temperature and humidity sensors used for the LCL estimation. The temporal evolution of the cloud fraction (blue line), the standard deviation of the cloud base on 30 minutes (dark line) and the difference between LCL and CBH (dark dashed line) are presented on the bottom panel. The vertical green dashed lines indicate the stratus onset and breakup times. The vertical red lines on the top and middle panels indicate the times of the visible and infrared cameras pictures presented on (b), (d) and (f) for IOP 8, IOP 14 and IOP 11, respectively. The times are 0814 and 1028 UTC on the 8 July, 0718 and 1214 UTC on the 27 July, and 0702 and 1036 UTC on the 18 July.





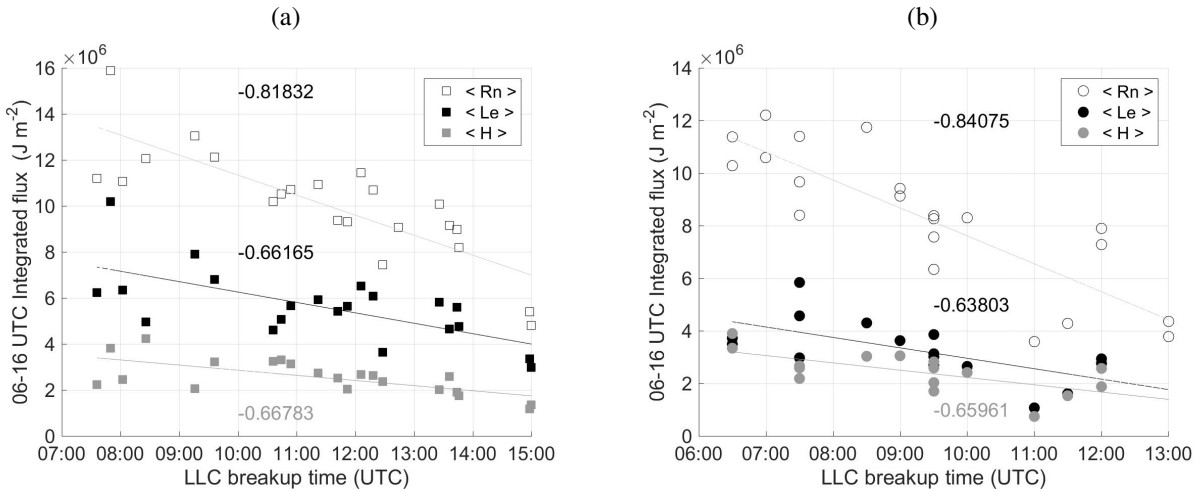

**Figure 8.** Integrated net radiation ($< Rn >$), latent ($< Le >$) and sensible ($< H >$) heat flux from 0600 UTC to 1600 UTC versus LLC breakup time at (a) Savè and (b) Kumasi. Linear regression are plotted and correlation coefficients are indicated.

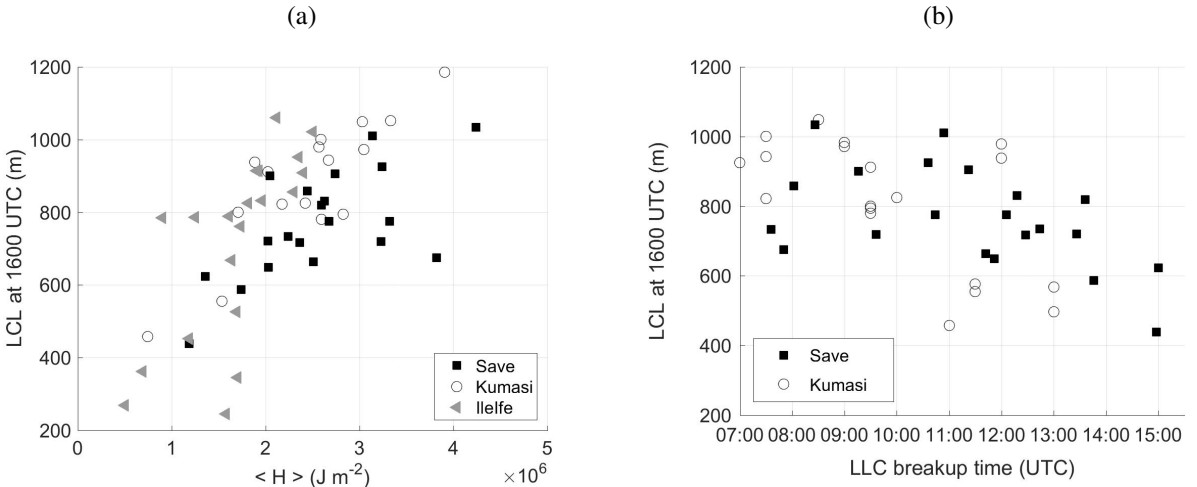

**Figure 9.** LCL at 1600 UTC against $< H >$ at the three sites (a). LCL at 1600 UTC against LLC breakup time at Savè and Kumasi (b). The LLC breakup time can not be estimated in Ile-Ife.