# Peer review of "Conceptual model of diurnal cycle of low-level stratiform clouds over southern West Africa"

_Atmospheric Chemistry and Physics, 2019_

## Referee Comment (RC1) · Anonymous Referee #1 · 30 Jul 2019

Measurements examining the low-level stratocumulus clouds over southern West Africa are summarized and integrated into a conceptual model that explains the general diurnal cycle and three different scenarios for the breakup of the clouds. The paper follows a clear, logical path that incorporates all available data processed with proper methods to support the proposed conceptual model. It is well-written with appropriate figures. Publication is recommended.

The only suggestion pertains to the interpretation of the horizontal advection and how it relates to the local cooling. Horizontal advection is just a transport of some atmospheric property by the motion of air, so it represents moving air around but it cannot actually cool the air to saturation. Since measurements are at a fixed point (Eulerian), it is necessary to include horizontal advection. However, the cooling rate following a parcel

(Lagrangian perspective) is driven by diabatic processes or through adiabatic cooling associated with upward vertical motion. Those are the processes that are actually responsible for cooling air to saturation. So, it would be good to acknowledge and discuss this caveat since it is stated several times that horizontal advection contributes to 50% of the total cooling. It might be helpful to refer to the cooling as 'local cooling' to reflect the idea that cooling is at a fixed point.

---

## Referee Comment (RC2) · Anonymous Referee #2 · 14 Oct 2019

"Conceptual model of diurnal cycle of stratiform low-level clouds over southern West Africa" by Lohou et al.

Summary: This paper presents a conceptual model for the diurnal cycle of low-level stratiform clouds over southern West Africa during the summer monsoon. This paper is relevant to a special issue on the DACCIWA field campaign and is worthy of publication in a special issue of ACP on the subject. For a conceptual model paper its presentation must be perfect and most of the comments below pertain to this point. These are all minor points, though there are several of them.

Major Comments:

1) Important to have consistency throughout. In the abstract the phrase "low-level

stratiform clouds" occurs while in the title it is written as "stratiform low-level clouds". Because "low-level stratiform clouds" is used throughout the manuscript, a more consistent title would be "Conceptual model of the diurnal cycle of low-level stratiform clouds over southern West Africa".

2) A better initialism than "LLC" for "low-level stratiform clouds" would be "LLSC", and for three reasons. The first reason is that "stratiform" is an integral adjective in the clouds studied in this manuscript. A second reason is that to sound out in one's mind "LLC" the word "stratiform" does not occur, so only "low-level clouds" appear, but one knows that "stratiform" is important so one has to insert the word into the "LLC" initialism, making it just a tad bit harder to read the paper. The third, and final, reason is that "LLC" is too close to "LCL", inviting confusion. In one place (at least) the authors mix them up too. Whoops!

3) Figure 2 must be perfect, yet it has a number of things that need to be improved. In the figure caption the words "The greenish rectangles and triangles" occur. First, the figure looks to contain more blues than greens. Second, the horizontal advection symbol is neither a rectangle nor a triangle. As a result, this part of the caption is not helpful and needs to be cleaned up.

Second, the sentence in the caption that reads "The white dashed curves indicate the lifting condensation level (LCL)." needs more explanation in the caption than provided. It is not until Pages 13-14 that they are discussed. Something like "Each of the three LCL curves represents one scenario (of three) of CBL development found during the DACCIWA field campaign (Section 3.2.3)." should be added to the caption. Also, "Scenario 1", "Scenario 2", and "Scenario 3" should be used as labels for the dashed lines in the figure. See marked up manuscript.

Each little rotation symbol in columns 2 and 3 of Figure 2 needs a precise meaning. For example, based on the text, the clockwise rotating symbols in the middle of the stratus layer in column 3 would seem to indicate downward mixing caused by cloud-

top cooling. If this is correct, the symbols make sense; if this is not correct, this is an example of of these symbols causing confusion. The clockwise rotating symbols in the sub-cloud layer of column 3 seem to indicate downward mixing caused by the jet shear. Correct? If so, what does the counter-clockwise rotation symbol mean near the surface along the low level jet curve in column 2? Right above it is a clockwise rotating symbol so the juxtaposition of the two is confusing. Moreover, in a conceptual model the juxtaposition of two such symbols should be made perfectly clear in the text. In column 2 there is also a counter-clockwise rotation symbol at $Z^* = 1.5$. What does this mean? According to its location along the low level jet curve, it might be taken to mean upward mixing due to shear. But in column 3 the rotation symbol at about $Z^* = 2.2$ is clockwise, perhaps indicating downward mixing due to cloud-top cooling. If so, why is it placed right on top of the low level jet curve in column 3?

For Figure 2 to be most effective, every drop of ink on it needs a clear purpose and one that is described in the text and easily remembered.

4) Figures 2 and 3 work really well together. Fun to read about them. The text on Pages 10 and 11 was a bit ambiguous in making perfectly clear that the curves in Figures 4 and 5 were averages over different phases. Or perhaps stated differently, it is disconcerting to see figures based on averages over different phases when the point of the paper is a conceptual model of the distinctness of the phases themselves. Figures 4 and 5 must contain averages over the four distinct phases to be most effective, even if some of the averages from one phase to the next are similar. All in all, Figures 4 and 5 were not intellectually satisfying, especially in comparison to Figures 2 and 3.

Minor Details:

0) A marked-up manuscript is being returned to the authors. It may contain detailed comments that they may find useful. The handwriting on the manuscript is not always so good, which is unfortunate. My apologies!

The more important points in the marked-up manuscript now follow.

1) The first paragraph on Page 5 (Lines 1-11) is not relevant to the main story of the manuscript and can be removed. The sentence on Lines 6-8 was relevant and can be moved to Page 16, Lines 20-25, as additional factors to consider.

2) The legends of color dots in Figures 1a and 1b need labels.

3) "Q" is not defined in Eq. 1 on Page 10.

4) The authors need to rethink their use of "the day after." This finally became clear on the bottom of Page 11. Here, it is stated that stratus occurs from 2200 UTC on day 1 out to 0500 UTC on day 2. In this case "the day after" on Line 26 would mean there is a day 3. Maybe just using "day 1" and "day 2" would be simpler and more exact.

5) The change in significant digits on Page 11, Lines 8-21, was a bit jarring.

6) Not sure that the sentence on Page 13, Lines 13-15, means exactly the same thing as the sentence on Lines 3-5 of the Figure 7 caption.

7) Page 13, Line 12: Never seen the word "summit" in this context. How about "radar cloud top"? If not "radar cloud top" this word needs to be defined.

8) The words "most likely" and "can" showed up a bit on Page 16. These are weak words in this context because they imply a weaker conceptual model. They should be removed from the manuscript in some way.

9) Figure 3: The x-axis tick marks must represent hours. So, the x-axis labels should have the units of time in them. Perhaps "(hr)"?

10) Figures 4a and 4b: The y-axis tick mark labels should have the same number of significant digits.

The two "RES" lines in Figure 4b should have the same color because they both represent residual curves. To distinguish between them, one can add (TURB) to "RES" in the left column legend and (TURB+RAD) to "RES" in the right column legend.

And again, one set of curves for each phase would be much more effective.

11) Figure 7 visible and infrared images: The features in the visible and infrared images do not seem to line up. Is the because their fields of view and their orientations do not line up? The infrared images seem to have a color bar at the bottom of the images whereas the visible images have no grey scale. It would be helpful to have visible and infrared images with the same fields of view and the same orientation relative to north (top), south (bottom), west (left), and east (right) relative to the page. Also, some sort of grey and color bars with labels would be helpful. These changes would make the images work better together.

By "incertitude", is "uncertainty" meant"? If so, why not use "uncertainty"?

Please also note the supplement to this comment:

[revised manuscript text omitted]

---

## Referee Comment (RC3) · Anonymous Referee #3 · 6 Nov 2019

Review of the article titled "Conceptual model of diurnal cycle of stratiform low-level clouds over southern west Africa" by Fabienne Lohou and coauthors for publication in the Atmospheric Chemistry and Physics.

The authors have used data collected during the DACCIWA field campaign along the African coast to characterize the diurnal cycle of boundary layer and clouds in the region. The article is essentially a summary of several other articles that have used data collected from this field campaign, as the data were collected at several sites. Based on the limited number of observations, the authors have classified the diurnal cycle into four phases, i) marine intrusion, ii) low level jet, iii) stratus, and iv) convective. They have then calculated the energy budgets for the first two phases, and have done some diagnostics for the second two. The idea of classifying the evolution in four

phases, doing budgets, and probing coupling with the surface are useful and worthy. However, the article falls short in several ways. I believe that the issues listed below need to be addressed for the article to be relevant for wider community. The main issue is lack of concrete motivation for and conclusions drawn from some of the analysis. Hence, I recommend this article for major revisions.

Major Issues

In Figure 6 you have shown the scatter plot of degree to decoupling (Bretherton and Wyant, 1997; Jones et al. 2011) and the bulk Richardson number. Maybe you can show the same plot colored by the different phases of the boundary layer mentioned in table 1, or color code them with the cloud fraction. This will show if the stratus clouds are coupled to the surface or not and whether the shallow cumulus form due to surface heating or shear (Zhu et al. 2001 JAS). Consistent with previous studies mentioned above, maybe you can calculate the ratio of cloud top cooling and surface heating, and contrast that with decoupling index.

Figure 8 and associated text: Please mention some previous studies that have shown any relationship between the "integrated flux" and "LLC breakup time". It is unclear what you mean by integrated flux and how it was calculated. Also, why do you choose to calculate integrated flux rather than the average flux itself like that has been done by numerous studies? This is especially crucial as the measurements are made in a Eulerian setting and there is no way of knowing the "history" of the parcels. Also, how do you define the LLC breakup time, what is the objective criteria for determining it? Thanks.

Similarly, in Figure 9 you have shown a scatter plot between surface flux and LCL. Under high surface flux conditions due to stronger mixing, the mixed layers are deeper than those under weaker surface flux conditions. Not sure if this is something worth showcasing in a paper. However, it is puzzling to see that in Figure 9b that a lower LCL results in a later breakup. Do you have any physical explanation of this? The text only

describes the figure without drawing any conclusions. Thanks.

Line 17 on page 12 says that "the stratus reduces the NLLJ strength because of turbulent mixing in the cloud layer". It is unclear to me where in the manuscript have you have shown this, or any other manuscripts that have shown this? The Stratus clouds can surely modify the boundary layer turbulence, but the LLJ is a meso-gamma scale phenomenon (∼200 km), much greater than the typical scales of stratocumulus clouds. Please show evidence of this or remove the sentence.

The moisture advection can bring saturation at the top of the boundary layer causing cooling. Thereby forming clouds and not needing surface moisture. This seems to be the case scenario 2 (Figure 7c and 7e). In other cases, it seems that the clouds are coupled to the surface even at night (Figure 7a). It is unclear to me how you objectively defined the three scenarios.

Page 6, Line 14: you calculated the stratus cloud fraction by using the cloud base height values below 600 m, however in schematic shown in Figure 2 it is apparent that stratus clouds can exist with bases above 1 km. Would it be possible to re-do the figure with bases below 1 or 1.5 km? this also contradicts the text on Page 13, line 13 that uses 1 km threshold.

Minor Issues Abstract, page 1 line 14: Insert "surface" before buoyancy. Thanks. Page 2 line 1: I would say "form" rather than "appear". Page 2, line 15: Insert "the" before "daytime". Figure 1: The thick circle denoting it to be a super-site at Ile-Ife doesn't line up with the other filled circles in both panels. Page 6 line 25: use "at which" rather than "when". Page 7 text: It doesn't have line numbers so difficult to point out, but it needs to be revamped for grammar. Thanks. The UTC time is same as Local time. This needs to be mentioned somewhere in the text. Thanks. Table 1: Would it be possible for you to mention in a column the average and standard deviation of surface fluxes, cloud fraction, cloud top height and cloud base height for each phases? Thanks. Page 13, Line 7: Change to "between this level and the bases of cumulus clouds forming

in the afternoon". Page 13 Line 12: Please change "summit" to "cloud top". It is also elsewhere like caption of Figure 7. Figure 3: There are no "red numbers" in the figure. Figure 4 and 5: Please mention the significance of the dashed line.

---

## Author Comment (AC1) · 27 Nov 2019

**Interactive comment on "Conceptual model of diurnal cycle of stratiform low-level clouds over southern West Africa"**

**Response to reviewer 1**

**Dear reviewer 1,**

**We thank the reviewer for his/her suggestion which led to clarify the contribution of the different terms to the total local cooling.**

Measurements examining the low-level stratocumulus clouds over southern West Africa are summarized and integrated into a conceptual model that explains the general diurnal cycle and three different scenarios for the breakup of the clouds. The paper follows a clear, logical path that incorporates all available data processed with proper methods to support the proposed conceptual model. It is well-written with appropriate figures. Publication is recommended. The only suggestion pertains to the interpretation of the horizontal advection and how it relates to the local cooling. Horizontal advection is just a transport of some atmospheric property by the motion of air, so it represents moving air around but it cannot actually cool the air to saturation. Since measurements are at a fixed point (Eulerian), it is necessary to include horizontal advection. However, the cooling rate following a parcel is driven by diabatic processes or through adiabatic cooling associated with upward vertical motion. Those are the processes that are actually responsible for cooling air to saturation. So, it would be good to acknowledge and discuss this caveat since it is stated several times that horizontal advection contributes to 50% of the total cooling. It might be helpful to refer to the cooling as 'local cooling' to reflect the idea that cooling is at a fixed point.

**We agree with reviewer that it would be clearer to precise "total local cooling" instead of "total cooling".  This has been changed at different places in the text. Also, the term TOT in equation 1 in the manuscript is now defined as local tendency of the potential temperature. We  think that a discussion about this would puzzle the reader since everything is clearly defined in equation (1). Equation (1) is an Eulerian equation and the term TOT (partial derivative of potential temperature with respect to time) is the cooling at fixed point which is the local cooling. Because it is the local cooling, the equation includes the horizontal and vertical advection terms.**

---

## Author Comment (AC2) · 27 Nov 2019

**Interactive comment on "Conceptual model of diurnal cycle of stratiform low-level clouds over southern West Africa"**

Response to reviewer 2

Dear reviewer 2,

We thank the reviewer for his/her valuable and constructive suggestions, which led to significant improvements of the quality of our manuscript. Below we detailed how his/her comments are addressed in the revised version of the manuscript. The major corrections made in the manuscript and cited in this document appear in italic. All the typographic, orthographic, syntactic corrections detailed in the supplement were more than appreciated! We hope we included all of them correctly in the text; however, we must admit that we could not read three of them (see at the end of the response).

Major Comments:
1) Important to have consistency throughout. In the abstract the phrase "low-level stratiform clouds" occurs while in the title it is written as "stratiform low-level clouds". Because "low-level stratiform clouds" is used throughout the manuscript, a more consistent title would be "Conceptual model of the diurnal cycle of low-level stratiform clouds over southern West Africa".

We absolutely agree with the reviewer and the title has been changed.

2) A better initialism than "LLC" for "low-level stratiform clouds" would be "LLSC", and for three reasons. The first reason is that "stratiform" is an integral adjective in the clouds studied in this manuscript. A second reason is that to sound out in one's mind "LLC" the word "stratiform" does not occur, so only "low-level clouds" appear, but one knows that "stratiform" is important so one has to insert the word into the "LLC" initialism, making it just a tad bit harder to read the paper. The third, and final, reason is that "LLC" is too close to "LCL", inviting confusion. In one place (at least) the authors mix them up too. Whoops!

We also agree with the reviewer; we actually used the initialism LLSC in one of the paper submitted at the same ACP special issue (Dione et al), and it make sens to be consistent with it. This will help the reader AND the authors not to mix LLC and LCL up.

3) Figure 2 must be perfect, yet it has a number of things that need to be improved.
In the figure caption the words "The greenish rectangles and triangles" occur. First, the figure looks to contain more blues than greens.
"Greenish" has been replaced by "Blue" in the legend.
Second, the horizontal advection symbol is neither a rectangle nor a triangle. As a result, this part of the caption is not helpful and needs to be cleaned up.
The shapes have been chosen according to the contribution of each term to the total local cooling, as a function of height. The radiative term is symbolized by a rectangle because it is constant with height. The turbulent term is symbolized by a triangle because it decreases with height. The horizontal advection is maximum at $Z^*=1$ but is half of its maximum value at surface and at $Z^* = 2$, so it is symbolized neither by a triangle nor by a rectangle, but by an appropriate shape corresponding to this profile. A sentence about the choice of the shape is included now in the legend (see below).

Second, the sentence in the caption that reads "The white dashed curves indicate the lifting condensation level (LCL)." needs more explanation in the caption than provided. It is not until Pages 13-14 that they are discussed. Something like "Each of the three LCL curves represents one scenario (of three) of CBL development found during the DACCIWA field campaign (Section 3.2.3)." should be added to the caption. Also, "Scenario 1", "Scenario 2", and "Scenario 3" should be used as labels for the dashed lines in the figure. See marked up manuscript.

Scenario 1, 2 and 3 have been added in the figure.

Each little rotation symbol in columns 2 and 3 of Figure 2 needs a precise meaning.
For example, based on the text, the clockwise rotating symbols in the middle of the
stratus layer in column 3 would seem to indicate downward mixing caused by cloud-top cooling. If this is correct, the symbols make sense; if this is not correct, this is an example of of these symbols causing confusion. The clockwise rotating symbols in the sub-cloud layer of column 3 seem to indicate downward mixing caused by the jet shear. Correct? If so, what does the counter-clockwise rotation symbol mean near the surface along the low level jet curve in column 2? Right above it is a clockwise rotating symbol so the juxtaposition of the two is confusing. Moreover, in a conceptual model the juxtaposition of two such symbols should be made perfectly clear in the text. In column 2 there is also a counter-clockwise rotation symbol at $Z^* = 1.5$. What does this mean? According to its location along the low level jet curve, it might be taken to mean upward mixing due to shear. But in column 3 the rotation symbol at about $Z^* = 2.2$ is clockwise, perhaps indicating downward mixing due to cloud-top cooling. If so, why is it placed right on top of the low level jet curve in column 3?

We are so sorry to cause the reviewer such a trouble trying to find a logic in the rotation of the symbol. We did not intend to put signification to the clockwise or counter-clockwise rotation symbol; they were put at places where turbulent mixing is known to be important. The results of our analyses do not allow to discuss the downward or upward mixing at different levels in the cloud or subcloud layers. We agree with the reviewer that the two rotations, clockwise and counter-clockwise, were puzzling. Only one rotation for the symbol is now used in Figure 2 and its signification is defined in the legend.

For Figure 2 to be most effective, every drop of ink on it needs a clear purpose and one that is described in the text and easily remembered.

We agree with all the suggestions of the reviewer; the figure has been modified and the caption is now: *"Conceptual model for the LLSC diurnal cycle over southern West Africa. The height is normalized by the LLSC base height (CBH) when the clouds form. The grey shades represent the LLSC (stratus-fractus or stratus) or cumulus cloud. The three white dashed curves indicate the lifting condensation level (LCL). Each of them represents one scenario (of three) of CBL development found during the DACCIWA field campaign (Section 3.2.3). The dark blue lines reproduce the vertical profiles of the wind with an indication of its maximum value for each phase. The various blue symbols at the bottom represent the processes involved in the potential temperature tendency equation (their shape is adapted to the contribution of each term to the local total cooling, as a function of height) (Section 3.1.2). H stands for the surface sensible heat flux and is symbolized by an orange arrow during the convective phase. The white rotating arrows symbolize nocturnal dynamical turbulence either due to the radiative cooling at the cloud summit or to the wind shear in the NLLJ."*

4) Figures 2 and 3 work really well together. Fun to read about them. The text on Pages 10 and 11 was a bit ambiguous in making perfectly clear that the curves in Figures 4 and 5 were averages over different phases. Or perhaps stated differently, it is disconcerting to see figures based on averages over different phases when the point of the paper is a conceptual model of the distinctness of the phases themselves. Figures 4 and 5 must contain averages over the four distinct phases to be most effective, even if some of the averages from one phase to the next are similar. All in all, Figures 4 and 5 were not intellectually satisfying, especially in comparison to Figures 2 and 3.

The Figures 4 and 5 objective is to quantify the processes involved in the air saturation (so before the stratus phase), this is why we focus here on the stable and jet phases. Adding the same figures for the stratus phase would add the difficulty to estimate the latent heat release due to condensation. This can not be estimated since we do not have the liquid water content for low level cloud only (we do have LWP but some higher clouds are often present). The radiative term would also be very difficult to estimate.

We agree with the reviewer that these figures for the stable and jet phase separately would have been much more satisfactory. But we really wanted to show, in this paper, the results over the different DACCIWA sites as much as possible, in order to generalize this conceptual model, first established with data acquired at Savè (the most instrumented site), to other locations in southern west Africa. Separated analyses for Stable and Jet phases were done by Adler et al (2019) for Savè because the number of radiosoundings were sufficient for most of the IOPs (1 radiosoundings every 1.5hrs). The number of radiosoundings was not sufficient at Kumasi (1 radiosoundings every 3hrs) which prevents separated analyses for the two phases and the start of the Jet phase has not been established at this site by the fuzzy method detailed in Dione et al, 2019.

We made clear in the text page 10 and 11 that these two figures are averages over Stable and Jet phases. Page 10: *"Figure 4 shows averaged vertical profiles, over stable and jet phases, of specific humidity and temperature contributions to the total change in RH at the Savè and Kumasi sites. The height is normalized by the cloud base when the stratus form. The median value of the cloud-base at Savè and Kumasi are 227 m a.g.l. and 137 m a.g.l., respectively (Kalthoff et al., 2018). At Savè, the cooling causes at least 80 % of the RH increase. Adler et al. (2019), who analyzed the moistening for separated stable and jet phases at Savè, pointed out a weak moistening during the stable phase which is almost compensated by a drying occurring during the jet phase, leading to a 20 % contribution of the moisture to the increase in RH at the end of the two phases (Fig.4a)."*

Page 11: *" The vertical profiles of all these terms, averaged over stable and jet phases, are presented in Figure 5. If such an analysis was performed separately for stable and jet phases at the Savè site (Adler et al., 2019) (not shown), it is however not feasible at the Kumasi site because, among other reasons, the MI and NLLJ arrival times were not established, so that the start of the jet phase is unkonwn."*

Minor Details:
0) A marked-up manuscript is being returned to the authors. It may contain detailed comments that they may find useful. The handwriting on the manuscript is not always so good, which is unfortunate. My apologies! The more important points in the marked-up manuscript now follow.

1) The first paragraph on Page 5 (Lines 1-11) is not relevant to the main story of the manuscript and can be removed. The sentence on Lines 6-8 was relevant and can be moved to Page 16, Lines 20-25, as additional factors to consider.

We agree that the paragraph does not respect the logic of the introduction. So we moved it in the conclusion (penultimate paragraph) and removed two of the references which were not directly linked to LLSC studies. The paragraph is now : *"A factor to also consider in the study of LLSC diurnal cycle is the aerosol effect in the context of rapid and significant socioeconomic changes that are happening in the southern West Africa (Knippertzet al., 2011). Deetz et al., 2018 performed highly resolved process study simulations for 2–3 July 2016 with COSMO-ART to assess the aerosol direct and indirect effect on meteorological conditions over southern West Africa. They find that MI and stratus-to-cumulus transition are highly susceptible to the aerosol direct effect, leading to a spatial shift of the MI front and a temporal shift of the stratus-to-cumulus transition with changes in the aerosol amount. On the other hand, aircraft measurements of aerosols and clouds over southern West Africa during the 2016 summer monsoon show pollution and polluted clouds across the whole region (Taylor et al., 2019, Haslett et al., 2019)"*.

2) The legends of color dots in Figures 1a and 1b need labels.

The labels have been added to the legend of color dots.

3) "Q" is not defined in Eq. 1 on Page 10.
The definition of Q has been added on page 10: *" where u, v, w are the wind components, $\rho$ the mean air density, Q the net radiation flux,..."*.

4) The authors need to rethink their use of "the day after." This finally became clear on the bottom of Page 11. Here, it is stated that stratus occurs from 2200 UTC on day 1 out to 0500 UTC on day 2. In this case "the day after" on Line 26 would mean there is a day 3. Maybe just using "day 1" and "day 2" would be simpler and more exact.
We thank the reviewer for taking note of this mistake. The sentence has been modified as follow: *"The stratus appears after 2200 UTC on day D and, for most of the cases, between 0000 and 0500 UTC on day D+1. The breakup occurs before 1500 UTC on day D+1."*
In the abstract, the sentence *"The LLSC, inaccurately represented in the climate and weather forecasts, form in the monsoon flow during the night and break up the day after,..."* is now : *"The LLSC, inaccurately represented in the climate and weather forecasts, form in the monsoon flow during the night and break up during the following morning or afternoon,..."*.

5) The change in significant digits on Page 11, Lines 8-21, was a bit jarring.

The significant digit is now uniform in the text.

6) Not sure that the sentence on Page 13, Lines 13-15, means exactly the same thing as the sentence on Lines 3-5 of the Figure 7 caption.

We do agree with the reviewer that the standard deviation of the cloud fraction within the time it is estimated (30 min) does not mean anything. The right information was given in the legend and the

sentence in the text is now: *"The half-hourly standard deviation of the cloud base, the cloud fraction (percentage of cloud base below 1000 m a.g.l. over 30 minutes) and the difference between cloud base height and LCL are indicated on the bottom panel."*

7) Page 13, Line 12: Never seen the word "summit" in this context. How about "radar cloud top"? If not "radar cloud top" this word needs to be defined.

"Cloud top" was what we meant. "*Summit*" has been replaced by "*top*" everywhere in the text.

8) The words "most likely" and "can" showed up a bit on Page 16. These are weak
words in this context because they imply a weaker conceptual model. They should be removed from the manuscript in some way.

Yes, this comes from a personal tendency not to be too peremptory. I removed this verbs which weaken the conclusion.

9) Figure 3: The x-axis tick marks must represent hours. So, the x-axis labels should have the units of time in them. Perhaps "(hr)"?

We agree with the reviewer that the unit is missing. The unit "(hr)" has been added to the x-axis label.

10) Figures 4a and 4b: The y-axis tick mark labels should have the same number of significant digits.

The two "RES" lines in Figure 4b should have the same color because they both represent residual curves. To distinguish between them, one can add (TURB) to "RES" in the left column legend and (TURB+RAD) to "RES" in the right column legend.

The figure has been modified as suggested.

And again, one set of curves for each phase would be much more effective.
See response to major comment #4.

11) Figure 7 visible and infrared images: The features in the visible and infrared images do not seem to line up. Is the because their fields of view and their orientations do not line up? The infrared images seem to have a color bar at the bottom of the images whereas the visible images have no grey scale. It would be helpful to have visible and infrared images with the same fields of view and the same orientation relative to north (top), south (bottom), west (left), and east (right) relative to the page. Also, some sort of grey and color bars with labels would be helpful. These changes would make the images work better together.

As indicated in the section 2, the visible images were full sky images whereas the aperture angles for the infrared camera were 43° times 32°, which corresponds to a 158 m times 114 m area at a height of 200 m. To make this information clearer in figure 7, and following the suggestion of the reviewer, we added a black rectangle in the visible images corresponding to the area of the infrared images.
Concerning the color scale, the infrared camera is a security camera. Up to now we failed to establish a simple law between the color scale and a brightness temperature, this is why we only use it in a

qualitative way. The color scale and the brightness temperature are now qualitatively linked in the legend; this information was sorely lacking. The legend is now: *"(a) 8 July (IOP 8), (c) 27 July (IOP 14), and (e) 18 July (IOP 11) illustrate scenarios 1, 2 and 3, respectively. The temporal evolution of the cloud base height (red dots) and cloud summit (grey dots) and LCL with its uncertainty (black line with grey shading) are presented in the top panels. The LCL uncertainty is based on the uncertainties of the temperature and humidity sensors used for the LCL estimation. The temporal evolution of the cloud fraction (blue line), the standard deviation of the cloud base over 30 minutes (black line), and the difference between LCL and CBH (black dashed line) are presented in the bottom panels. The vertical green dashed lines indicate the stratus onset and breakup times. The vertical red lines indicate the times of the visible and infrared cameras pictures presented on (b), (d), and (f) for IOP 8, IOP 14 , and IOP 11, respectively. The times are 0814 and 1028 UTC on 8 July, 0718 and 1214 UTC on 27 July, and 0702 and 1036 UTC on 18 July. The color scale for the infrared images (indicated at the bottom of the images) ranges from blue for colder brightness temperatures, to white for the warmer brightness temperatures. The black rectangle on the visible images indicate the area corresponding to the infrared images. The white dot in the infrared images, at 1214 UTC on 27 July and 1036 UTC on 18 July, is due to the sun and is located at the center of the solar disk."*

By "incertitude", is "uncertainty" meant"? If so, why not use "uncertainty"?

Sorry we meant uncertainty. The changes have been done.

About the corrections in the supplement:
P5 l12 and P15 l12: The reviewer suggests "the most important details" instead of "the most important lessons". "most important" and "details" seemed contradictory to us, so we did not make the change.

P12 l3-5 and P15 l15-20: we could not read the reviewer's comments in the left margin.

---

## Author Comment (AC3) · 27 Nov 2019

Figure 6 is built with data acquired during the stratus phase; the objective was to analyze the coupling during that phase before the start of the surface thermal convection (the coupling during the convective phases is out of the scope of this paper). During that phase the cloud fraction is 100%. We do agree with the reviewer that this should clearly be indicated in the text and in the legend and those have been modified.

The sentence P12-l18 is now: *"Figure 6, where the bulk Richardson number in the subcloud layer is plotted against the difference between the CBH and the LCL estimated from the radiosondes **launched during the stratus phase** at Savè and at Kumasi."*
The legend is now: *"Bulk Richardson number ($Ri_b$) in the subcloud layer against height difference between cloud base height (CBH) and lifting condensation level (LCL) **estimated from the radiosondes launched during the stratus phase** at Savè and at Kumasi. Colors stand for the different IOPs."*

We do agree with the reviewer that the convective phase and especially the stratus to cumulus transition is also interesting in term of coupling. The convective phase is currently analyzed in details and the results will be submitted soon in an other manuscript. Unfortunately, it is quite difficult to estimate the cloud top cooling with the collected data as suggested by the reviewer, but we do try to give some clue on that point.

Figure 8 and associated text: Please mention some previous studies that have shown any relationship between the "integrated flux" and "LLC breakup time".
As suggested by the reviewer 3, we use now the averaged flux instead of the integrated flux (see response to following comment). The following sentence: "As expected, <Rn>, <LE>, and <H> are negatively correlated with the stratus breakup time with correlation coefficients below -0.64 for Savè and Kumasi (Fig. 8)." is now *"Figure 8 shows a negative correlation between <Rn>, <LE>, and <H> and the stratus breakup time with correlation coefficients below -0.64 for Savè and Kumasi."*

It is unclear what you mean by integrated flux and how it was calculated. Also, why do you choose to calculate integrated flux rather than the average flux itself like that has been done by numerous studies? This is especially crucial as the measurements are made in a Eulerian setting and there is no way of knowing the "history" of the parcels.

We agree with the reviewer that the averaged flux can be used in this study and all the figures are modified consequently. The way the integrated fluxes was calculated was indicated p14: "...the temporally integrated flux from 0600 UTC to 1600 UTC.". The unit was $Jm^{-2}$ as indicated in Figures 8 and 9. The averaged flux is the ratio of this integrated flux by the time duration (36 000 sec in our case - from 6 to 16 UTC). Consequently the scatter plot are perfectly similar with integrated or averages flux.

Also, how do you define the LLC breakup time, what is the objective criteria for determining it? Thanks.

The response to this comment is merged with the response to comment about cloud fraction (see below).

Similarly, in Figure 9 you have shown a scatter plot between surface flux and LCL. Under high surface flux conditions due to stronger mixing, the mixed layers are deeper than those under weaker surface flux conditions. Not sure if this is something worth showcasing in a paper.

We fully agree with the reviewer that this figure shows a very common results but we decided to keep this figure for three reasons:
1/ it is the only figure that presents some data from the third site Ile-Ife.
2/ it is part of the following argumentation (see comment below)
3/ very few measurements exit in this part of the world, so it seems important to show them when they exist even if the results are confirming some expected relationship.

However, it is puzzling to see that in Figure 9b that a lower LCL results in a later breakup. Do you have any physical explanation of this? The text only describes the figure without drawing any conclusions. Thanks.

This results is not that surprising considering the figures 8 and 9a. We improved the text to better emphasize this "*Finally, the link between the CBL development and the stratus breakup time is shown, for the Savè and Kumasi sites, in Figure 9b. Latter stratus breakup implies lower net radiation at surface (Fig. 8) and therefore weaker surface flux conditions (Fig. 9a) which lead to a lower vertical development of the CBL. The LCL is half, when LLSC breakup occurs after 1100 UTC, that compared to an LCL associated with early-morning LLSC breakup. The impacts of this on the moist convection during the afternoon need detailed investigations.*"

Line 17 on page 12 says that "the stratus reduces the NLLJ strength because of turbulent mixing in the cloud layer". It is unclear to me where in the manuscript have you have shown this, or any other manuscripts that have shown this? The Stratus clouds can surely modify the boundary layer turbulence, but the LLJ is a meso-gamma scale phenomenon (∼200 km), much greater than the typical scales of stratocumulus clouds. Please show evidence of this or remove the sentence.

We have no mean to show evidence of this except that the wind is reduced in the stratus layer as soon as this one forms. We tried to compare the vertical profile of the wind speed for cloudy nights and clear

nights but the clear nights are rare (1 or 2 in Savè). The conditions very special for these two cases with very weak NLLJ. We think that the turbulence in the cloud is able to induce the decrease of the wind speed because the NLLJ and the LLC do have the same horizontal scale. There are both meso-gamma phenomena. The stratus in southern west Africa extends over more than 800 000 km$^2$. The NLLJ can be mixed up in the stratus layer from the Guinean coast to Savè. However, we changed the sentence and suggest this as an explanation.

The following paragraph: *"As shown in Figure 3, the stratus and the NLLJ could interact in two ways: (1) the stratus may reduce the NLLJ strength because of the turbulent mixing in the cloudy layer, and (2) the turbulence below the NLLJ core modifies the conditions in the subcloud layer."* is now *"As shown in Figure 3, the stratus and the NLLJ could interact in two ways. First, the stratus could reduces the NLLJ strength because of the turbulent mixing in the cloudy layer. Such an effect of the stratus turbulent mixing on the meso-scale phenomenon that is the NLLJ, is possible because the stratus extend over more than 800 000 km$^2$ from the Guinean coast up to 10° N latitude. Secondly, the turbulence below the NLLJ core modifies the conditions in the subcloud layer."*

The moisture advection can bring saturation at the top of the boundary layer causing cooling. Thereby forming clouds and not needing surface moisture. This seems to be the case scenario 2 (Figure 7c and 7e). In other cases, it seems that the clouds are coupled to the surface even at night (Figure 7a). It is unclear to me how you objectively defined the three scenarios.

We do not understand what period or stage the reviewer is discussing in his comment. Concerning the stratus formation, we showed in Figure 4 that the change in relative humidity is due to the cooling and not to the moistening. The three scenarios have been objectively defined looking, case by case:

- the cloud base height relatively to the LCL along the stratus and convective phases; we interpret the departure between both in terms of coupling.
- the standard deviation of the cloud base; this discriminates stratus (low standard deviation) from cumulus clouds (high standard deviation).

All the stratus nights during the campaign at Savè follow one of the three scenarios described in the paper. We agree with the reviewer that we defined all the parameters shown in Figure 7 but they are not explicitly mentioned as criteria for the definition of the scenarios.

The following paragraph : *"The way in which the stratus layer and the surface are coupled (or not), as discussed in the previous section, plays a determining role on the breakup. Three scenarios have been observed at Savè, illustrated in Figure 7."* is now *" The way in which the stratus layer and the surface are coupled (or not), as discussed in the previous section, plays a determining role on the breakup. Based on CBH evolution relatively to LCL and CBH standard deviation along the stratus and convective phases, three scenarios have been defined at Savè, illustrated in Figure 7."*

Page 6, Line 14: you calculated the stratus cloud fraction by using the cloud base height values below 600 m, however in schematic shown in Figure 2 it is apparent that stratus clouds can exist with bases above 1 km. Would it be possible to re-do the figure with bases below 1 or 1.5 km? this also contradicts the text on Page 13, line 13 that uses 1 km threshold.

- We thank the reviewer for pointing this inconsistency in the text. The cloud fraction estimate using the cloud base below 600 m was first used by Adler et al, 2019. They used a 600 m height threshold because their focus was the stratus phase and the breakup time. Once the study has been extended to the convective phase and especially to the stratus-cumulus transition, the threshold has been increased to 1000 m because cumulus base can be higher than 600m.

- The stratus breakup time is determine as indicated p13-l8: *"A cloud fraction larger than 95 % is chosen as a criterium to determine the presence of stratus clouds above the supersite."*. We agree with the reviewer that this explanation should be also given p6 when the method for cloud fraction estimate is presented.

- The height in Figures 2 and 3 is normalized by the LLC base when the stratus forms. So these figures do not indicate directly the height of the base. However, we realized that the LLC base when the stratus forms is not quantified in the text.

Then, considering the three points above, several passages in the text have been modified:

P6, l4: *"A ceilometer was deployed at the Savè and Kumasi sites, continuously providing the cloud-base height (CBH). The LLSC fraction (Adler et al., 2019) was deduced from the percentage of CBHs measured below 1000 m. Adler et al., 2019 used a 600 m height threshold for the stratus phase analysis. This threshold is increased up to 1000 m in the present study, consistently with Dione at al. (2019), to allow the stratus to cumulus transition analysis during the convective phase. A cloud fraction larger than 95 % is chosen as a criterium to determine the presence of stratus clouds above the supersite, from which the stratus appearance and breakup times are deduced. An other method for theses two times estimate was use of the infrared cloud camera (Dione et al., 2019)."*

P9, l27: *"Figure 4 shows averaged vertical profiles, over stable and jet phases, of specific humidity and temperature contributions to the total change in RH at the Savè and Kumasi sites. The height is normalized by the cloud base when the stratus form. The median value of the cloud-base at Savè and Kumasi are 227 m a.g.l. and 137 m a.g.l., respectively (Kalthoff et al., 2018). At Savè, the cooling causes at least 80 % of the RH increase."*

Minor Issues
Abstract, page 1 line 14: Insert "surface" before buoyancy. Thanks. This has been modified.
Page 2 line 1: I would say "form" rather than "appear". This has been modified.
Page 2, line 15: Insert "the" before "daytime". This has been modified.
Figure 1: The thick circle denoting it to be a super-site at Ile-Ife doesn't line up with the other filled circles in both panels.
The filled circle for the permanent synoptic meteorologic station does not line up with the circle for the super site instrumented for the DACCIWA field experiment because the two sites are not exactly at the same location. We used the GPS coordinates.
Page 6 line 25: use "at which" rather than "when".
CBH stands for Cloud Base Height. "...cloud base height at which the low level cloud form..." does not seem correct to me. We wanted to say that, for the normalization, we use the height of the clouds when they form. We think that "CBH when the LLC form" is the good sentence to say this.
Page 7 text: It doesn't have line numbers so difficult to point out, but it needs to be revamped for grammar. Thanks.
We are sorry but we did not find in these 4 lines what needs to be corrected. Some minor changes have been made, suggested by reviewer 2
The UTC time is same as Local time. This needs to be mentioned somewhere in the text. Thanks.
We agree with the reviewer that difference between Local Solar time and UTC must be defined for each site. The following sentence has been added P, l: *"UTC and local solar time are only about 6 min,*

*10 min, and 18 min apart at Kumasi (-1.5601° E, 6.6796° N), Savè (2.4275° W, 8.0009°N), and Ile-Ife (4.5574°W, 7.5532°N), respectively. The results are henceforth presented according to UTC."*

Table 1: Would it be possible for you to mention in a column the average and standard deviation of surface fluxes, cloud fraction, cloud top height and cloud base height for each phases? Thanks.
We understand that the reviewer 2 ask these information for stratus and convective phases only and not for the stable and jet phases (since the stratus is not formed yet). All these information are sometimes difficult to provide; others are now included in the text or a reference is added :

- The surface flux are very low during the night (stratus phase) and furthermore very difficult to estimate for different reasons: (1) the sonic anemometer and the Licor hygrometer measurements are not systematic because of high level of relative humidity (~100%) or even some remaining droplets on the sensors, and (2) the hypothesis of homogeneity and stationary are very often not verified. This is why we started the temporal integration of the surface flux at 6 UTC. Concerning the convective phase, the average fluxes are given in Figure 8 and 9.
- There was a cloud radar in Savè, but the cloud summit was very often difficult to estimate from these measurements (Adler et al. , 2019). Furthermore this information was not available in Kumasi.
- As explained in a previous response, we added the median value of the LLC base during stratus phase at the two sites. The temporal evolution of the cloud base is presented in Kalthoff et al., 2018, and this information is now provided in the text. *"Kalthoff et al. (2018) present the temporal evolution of the distribution of the cloud base estimates by the ceilometer along the stratus and convective phases at the Savè and Kumasi sites."*

Page 13, Line 7: Change to "between this level and the bases of cumulus clouds forming in the afternoon". This has been modified.
Page 13 Line 12: Please change "summit" to "cloud top". It is also elsewhere like caption of Figure 7. This has been corrected.
Figure 3: There are no "red numbers" in the figure. This has been corrected.
Figure 4 and 5: Please mention the significance of the dashed line.
The following sentence has been added to the legend of figure 3, 4 and 5: *"The horizontal dashed line (Z\*=1) indicates the CBH when the stratus form."*

---

## Author Response (AR2)

Dear Pr. van den Heever,

We thank you for these 3 remarks and answered to each of them.

   (1) Reviewer 3 asks the following related to Figure 8 and the associated text: "Please mention some previous studies that have shown any relationship between the "integrated flux" and the "LLC breakup time". I assume that you did not include any references as you decided to use the averaged flux rather than the integrated flux. However, even when using the averaged flux I think that it would be useful to mention previous studies demonstrating this relationship.

We do not know any reference which shows a relationship between surface flux and LLSC breakup time; we should have mentioned this in the response to reviewer 3. If you know any, please let us know. However the relationship shown in Figure 8 seems to be reasonable as the stratus has a strong impact on the net radiation as already shown by Hartmann or Chen. In order to include this idea we added two sentences; the modified paragraphs are below with the new sentences in bold blue.

*"**The stratus clouds modify the surface energy balance (SEB) because they reduce the net shortwave radiation twice as much as they increase the net infrared radiation (Chen et al., 2000).** In order to investigate the effect of the LLSC on the SEB during DACCIWA campaign, the temporally averaged flux from 0600 UTC to 1600 UTC (indicated by < >) of the net radiation (Rn), the latent heat flux (Le), and the sensible heat flux (H) are calculated for 21 days for Savè and Kumasi and 20 days for Ile-Ife.*

*Figure 8 shows a negative correlation between <Rn>, <LE>, and <H> and the stratus breakup time, with correlation coefficients below -0.64 for Savè and Kumasi. The variability is certainly due to the day-to-day variation of soil moisture which is more important at the beginning of the campaign than at the end, when frequent rain events maintained an almost constant soil moisture. <Rn> is reduced by 25% and 50% when LLSC breakup occurs after 1300 UTC at Savè and at Kumasi, respectively. This difference may be due to LLSC macrophysical properties, like deeper clouds or larger liquid water path, but also to higher cloudy layers which also impact the net radiation. Unfortunately, the same plot cannot be provided for Ile-Ife where the breakup time of the stratus could not be determined. **From these results, one can deduce for the first time the error related to the SEB if the LLSC breakup time is inaccurately simulated by numerical models**."*

   (2) "Latter stratus breakup implies lower net radiation at surface (Fig. 8)" needs to be replaced with "Later stratus breakup .... at the surface (Fig. 8)" (not both "Later" and "the surface").

We apologize for these two mistakes in the same sentence. The sentence is now: "Later stratus breakup implies lower net radiation at the surface."

   (3) There are still a number of grammatical errors throughout the manuscript. Please will you work through the manuscript and try to eliminate as many of these as possible.

We corrected as many grammatical errors as possible.

In addition to these improvements, below is the list of the main modifications we did in the text:

1/ we removed the reference to Bessardon et al., since this article is not ready for publication yet.

2/ we replaced some "stratus" by "LLSC" in the text, including in some section or subsection titles, to be consistent. Example: "From LLSC onset to breakup" instead of "From stratus onset to breakup".

3/ A last sentence has been added to the following paragraph (in the conclusion) to better link it to the present study: "*A factor also to consider in the study of LLSC diurnal cycle is the aerosol effect in the context of rapid and significant socioeconomic changes that are happening in southern West Africa (Knippertz et al., 2011). Deetz et al. (2018) performed highly resolved process study simulations for 2–3 July 2016 with COSMO-ART to assess the aerosol direct and indirect effect on meteorological conditions over southern West Africa. They find that MI and stratus-to-cumulus transition are highly susceptible to the aerosol direct effect, leading to a spatial shift of the MI front and a temporal shift of the stratus-to-cumulus transition with changes in the aerosol amount. However, aircraft measurements of aerosols and clouds over southern West Africa during the 2016 summer monsoon show pollution and polluted clouds across the whole region (Taylor et al., (2019), Haslett et al. (2019)).* **The aerosol effect on LLSC diurnal cycle could not be investigated with ground-based measurements performed during DACCIWA campaign.**"

---

## Author Response (AR3)

Dear Pr. Van den Heever,

We corrected the last sentence as suggested...and the full paragraph is actually more logical like that. The sentence is now :

"Aircraft measurements of aerosols and clouds over southern West Africa during the 2016 summer monsoon show pollution and polluted clouds across the whole region (Taylor et al., (2019), Haslett et al. (2019)). However, the aerosol effect on LLSC diurnal cycle could not be investigated with ground-based measurements performed during DACCIWA campaign."

We really appreciated the time you spent on this review.

Best regards, Fabienne Lohou

---

## Author Response (AR4)

Dear Natascha Töpfer

I uploaded on ACP website all the files required for production.

All the best, Fabienne Lohou.